# Circuit-specific hippocampal ΔFosB underlies resilience to stress-induced social avoidance

Andrew L. Eagle 1,7, Claire E. Manning 1,7, Elizabeth S. Williams 1, Ryan M. Bastle2, Paula A. Gajewski1, Amber Garrison1, Alexis J. Wirtz1, Seda Akguen1, Katie Brandel-Ankrapp1, Wilson Endege3, Frederick M. Boyce3, Yoshinori N. Ohnishi4,5, Michelle Mazei-Robison1, Ian Maze2, Rachel L. Neve6 & Alfred J. Robison 1✉

Chronic stress is a key risk factor for mood disorders like depression, but the stress-induced changes in brain circuit function and gene expression underlying depression symptoms are not completely understood, hindering development of novel treatments. Because of its projections to brain regions regulating reward and anxiety, the ventral hippocampus is uniquely poised to translate the experience of stress into altered brain function and pathological mood, though the cellular and molecular mechanisms of this process are not fully understood. Here, we use a novel method of circuit-specific gene editing to show that the transcription factor ΔFosB drives projection-specific activity of ventral hippocampus glutamatergic neurons causing behaviorally diverse responses to stress. We establish molecular, cellular, and circuit-level mechanisms for depression- and anxiety-like behavior in response to stress and use circuit-specific gene expression profiling to uncover novel downstream targets as potential sites of therapeutic intervention in depression.

[1] Department of Physiology, Michigan State University, East Lansing, MI, USA. [2] Department of Neuroscience, Icahn School of Medicine, Mount Sinai, New York, NY, USA. [3] Department of Neurology, Massachusetts General Hospital, Cambridge, MA, USA. [4] Department of Pharmacology, Kurume University School of Medicine, Kurume, Fukuoka, Japan. [5] Department of Medical Biophysics and Radiation Biology, Faculty of Medical Sciences, Kyushu University, Fukuoka, Japan. [6] Gene Technology Transfer Core, Massachusetts General Hospital, Cambridge, MA, USA. [7]These authors contributed equally: Andrew L. Eagle, Claire E. Manning. ✉email: juice5181@gmail.com

The ventral hippocampus (vHPC) is uniquely positioned to regulate emotional responses to stress that may underlie neuropsychiatric diseases such as depression[1,2]. Glutamatergic vHPC neurons project to regions important in stress susceptibility and mood, including the nucleus accumbens (NAc) and basolateral amygdala (BLA), and activity of NAc-projecting vHPC neurons mediates reward behavior[3] and susceptibility to stress-induced social avoidance[4]. We recently demonstrated that hyperexcitability of NAc-projecting vHPC neurons also underlies female-specific susceptibility to anhedonia in the subchronic variable stress model[5], and others have recently shown that activity in this circuit is broadly predictive of stress-induced susceptibility to anxiety and social withdrawal behaviors[6]. However, we do not have a clear understanding of the molecular mechanisms in the vHPC driving its modulation of these target regions in depressive and anxiety disorders. Stress alters gene expression in vHPC neurons[7–9], but the molecular mechanisms underlying circuit-specific vHPC gene expression and activity in response to stress are poorly understood.

ΔFosB is a remarkably stable transcription factor[10,11] found throughout the brain and induced by chronic neuronal activity[12]. ΔFosB has a well-established role in NAc in mediating stress susceptibility[13–15] and regulates synaptic and intrinsic properties of NAc medium spiny neurons[16]. It is also necessary in dorsal hippocampus (dHPC) for learning[17], and regulates the excitability of dHPC CA1 neurons[18] as well as hippocampal neurogenesis[19,20]. ΔFosB is induced throughout HPC by stress and antidepressant treatment[13,21–23], and vHPC CA3 ΔFosB is critical for the prophylactic effects of ketamine on stress responses[23]. This indicates that ΔFosB is a critical modulator of vHPC function and may orchestrate long-term alterations in gene expression underlying depressive and anxiety disorders. Here, we use circuit-specific CRISPR gene editing to reveal a previously unknown role for ΔFosB in vHPC neurons projecting to NAc in resilience to social defeat stress-induced social avoidance. Furthermore, we show that ΔFosB regulates the excitability of this circuit and we identify potential downstream gene targets in this circuit that may underlie stress resilience.

## Results

To investigate the induction of ΔFosB in vHPC by stress, we exposed male C57Bl6/J mice to chronic social defeat stress (CSDS, Fig. S1a), which produces a variety of depressive-like symptoms, including a social avoidant phenotype in stress-susceptible mice[24–26]. Stress increased the number of ΔFosB+ dentate gyrus neurons in vHPC (Fig. S1b, c). Expanding upon our previous findings in the dHPC[21,22], we found that ΔFosB was also induced in all subregions of the vHPC following repeated treatment with the antidepressant fluoxetine (Fig. S1d, e), a selective serotonin reuptake inhibitor. Thus, ΔFosB is induced in vHPC by both stress and antidepressants, which strongly suggests that stress-induction of ΔFosB in vHPC is a compensatory response to counteract the effects of stress, i.e., mediating stress resilience as it does in NAc[14].

Because the excitability of afferent projections from vHPC CA1 extending to NAc mediate CSDS susceptibility[4] and ΔFosB regulates excitability of dHPC CA1 neurons[18], we set out to investigate the role of vHPC projection neuron ΔFosB in CSDS susceptibility to social avoidance. To label vHPC-NAc neurons, we utilized a mouse line expressing Cre-dependent GFP-L10a ribosomal subunit fusion protein (*Rosa26*$^{eGFP-L10a}$), and injected into NAc a persistent, retrograde viral vector expressing Cre (HSV-hEf1α-Cre). Three weeks after surgery, GFP expression can be observed in ventral CA1 of vHPC (vCA1) and in ventral subiculum (vSub) (Fig. S2). Mice were then exposed to CSDS and

immunostained for ΔFosB. We found that CSDS induced ΔFosB in labeled vHPC-NAc projections (Fig. 1a, b). To provide additional quantitative assessment, we used the same mouse model but took bilateral punches containing GFP-labeled vHPC-NAc projections and processed for circuit-specific translating ribosome affinity purification (TRAP) to enrich for actively translating *FosB* and *ΔFosB* mRNA in vHPC-NAc neurons. CSDS produced a greater than fourfold change in these transcripts (Fig. S1f). These findings conclusively demonstrate that chronic stress increases ΔFosB expression in vHPC neurons projecting to NAc. Our next question then was to determine whether ΔFosB in this circuit regulates stress phenotypes.

We first probed whether ΔFosB-mediated transcriptional regulation in vHPC is critical in resilience to stress-induced social avoidance. We virally overexpressed ΔJunD, a transcriptional inhibitor of ΔFosB[14], in either the vHPC or dHPC (Fig. 1c) and exposed mice to a subthreshold microdefeat stress (Fig. 1d) which produces a social avoidant phenotype only in mice sensitized to stress[24,25]. Transcriptional inhibition of ΔFosB in vHPC reduced social interaction (SI) (Fig. 1e): ΔJunD mice spent less time interacting with a social target. Conversely, inhibition of ΔFosB in dHPC did not produce a stress-susceptible phenotype (Fig. 1f). Furthermore, vHPC ΔFosB inhibition but did not affect baseline (unstressed) anxiety or locomotor activity (Fig. S3). Thus, ΔFosB function in vHPC is necessary for resilience to stress, but its circuit-specific role remained unknown.

Because tools to test the circuit-specific role of an individual gene have not previously been described, we developed an innovative CRISPR-Cas9 viral toolset to manipulate ΔFosB expression in a circuit-specific manner (Figs. S4, S5; Suppl. Methods). Using CRISPR, we developed and screened guide RNAs (gRNA) specific to the *FosB* gene, which encodes ΔFosB, and tested these in cultured mouse neuroblastoma (Neuro2A) cells. Transfection of cells with Cas9 and gRNA specific to exon I of the *FosB* gene reduced ΔFosB expression (Fig. S4). To confirm this in vivo, an HSV expressing both Cas9 and the gRNA was infused into dHPC of adult mice. Infected dHPC neurons had significantly reduced ΔFosB expression compared to GFP only controls (Fig. S4d, e), indicating that the CRISPR-Cas9 vector successfully mutated the *FosB gene* and silenced expression of ΔFosB protein (*FosB* KO). Previously, we have shown that ΔFosB transcriptional inhibition (via ΔJunD) in dHPC impairs learning and memory, including novel object recognition[17]. In keeping with our previous findings, infusion of CRISPR *FosB* KO vector in dHPC also impaired recognition memory (Fig. S4f).

To produce a circuit-specific tool to efficiently target ΔFosB in only vHPC afferent neurons, we split the CRISPR constructs (Cas9 and gRNA) into two separate viral vectors to allow a circuit-specific intersection approach (Fig. S5a). Thus, we injected a retrograde viral vector into a projection target region and a locally expressing vector into the somatic region. Retrograde virus encoded a Cre-dependent Cas9 (HSV-hEf1α-LS1L-Cas9) and was injected into NAc; mice were allowed to recover for 3 weeks for maximal retrograde expression. We then injected into the vCA1/vSub an HSV that locally expresses Cre and *FosB* gRNA (HSV-IE4/5-TB-gRNA-eYFP-CMV-IRES-Cre), or control HSV (HSV-IE4/5-TB-eYFP-CMV-IRES-Cre). Similar to our finding with the single-vector CRISPR strategy, the dual-vector approach specifically reduces the expression of ΔFosB in co-labeled GFP and Cas9 vHPC cells, i.e., in NAc-projecting vHPC neurons (Fig. S5b–d).

Our tool produced a circuit-specific, CRISPR-mediated silencing of the *FosB* gene (*FosB* KO) leading to reduced ΔFosB protein expression in vHPC cells projecting to NAc, allowing us to interrogate the circuit-specific effects of ΔFosB on stress susceptibility. Adult mice that received *FosB* KO in the vHPC-NAc circuit (Fig. 2a) were subjected to CSDS and assayed for social

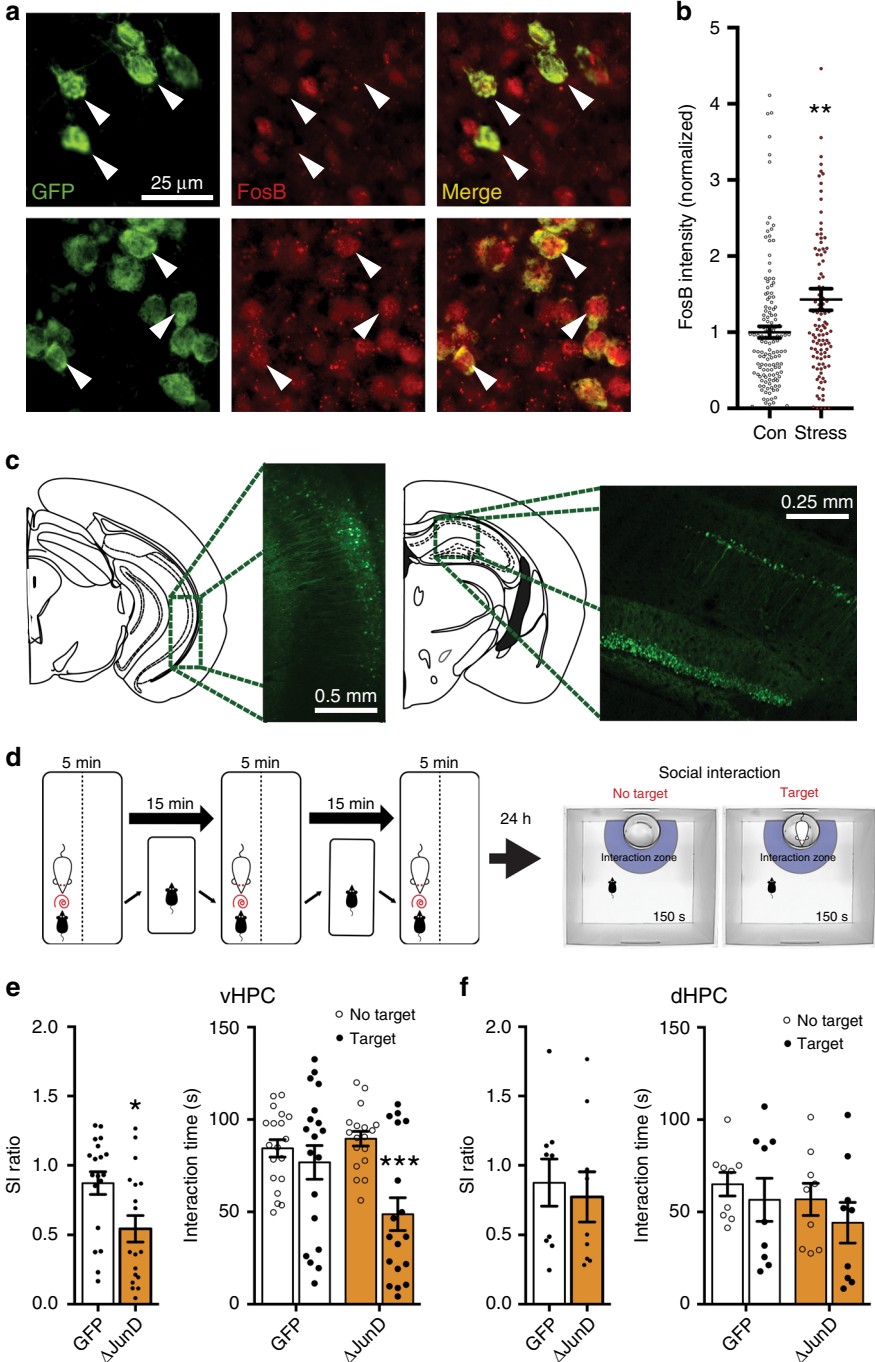

**Fig. 1 ΔFosB expression in the ventral hippocampus is necessary for CSDS resilience. a** Representative images of vHPC CA1 coronal sections (×40) showing immunofluorescent labeling of NAc-projecting neurons expressing GFP (left), ΔFosB (red, middle) and merge (right). Stressed mice (bottom panels; $n = 109$ cells) show increased ΔFosB signal in GFP-positive cells compared to controls (top panels; $n = 147$ cells), as indicated by white arrows and quantified in (**b**). **P = 0.0048 (independent samples $t$-test compared to Control; 4 data points are outside the axis). **c** Representative figures and coronal sections showing viral-mediated GFP expression in ventral and dorsal HPC. **d** Experimental design for subthreshold defeat and social interaction (SI) test. **e** ΔJunD inhibition of ΔFosB in vHPC reduced SI ratio (left) and decreased investigation time of the social target (right). *P = 0.0125, ***P < 0.0001 ($n = 19$ GFP, $n = 18$ ΔJunD; SI ratio: independent samples $t$-test compared to GFP; investigation time: two-way mixed ANOVA with Holm–Sidak post-test GFP No Target vs Target). **f** ΔJunD expression in dHPC did not affect stress-induced social avoidance ($n = 9$ mice/group). All graphs are represented as mean ± SEM.

avoidance. *FosB* KO in vHPC-NAc enhanced stress-induced social avoidance (Fig. 2b) and caused a small decrease in locomotor activity (Fig. S6a). These findings suggest that ΔFosB likely regulates locomotor activity and stress susceptibility through the vHPC-NAc circuit. This is consistent with previous reports showing that vHPC afferents to NAc regulate locomotor activity[27,28]. It is critical to note that this manipulation of *FosB* in vHPC-NAc neurons did not affect passive avoidance learning (Fig. 2c) or baseline anxiety (Fig. 2d), behaviors not associated with vHPC-NAc projections. We also observed stress effects in allodynia and anxiety, but vHPC-NAc FosB KO did not alter these behaviors (Fig. S6c–f). Therefore, ΔFosB expression in

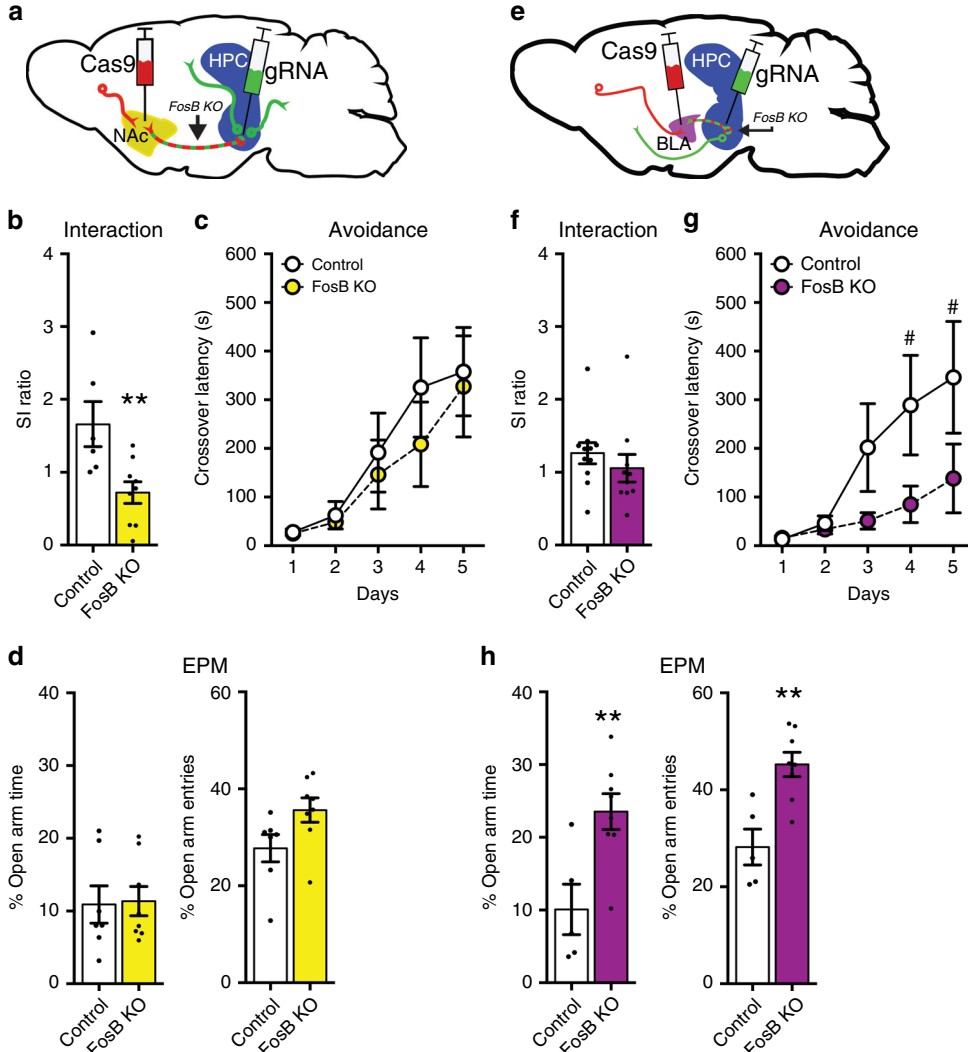

**Fig. 2 Circuit-specific silencing of *FosB* gene in ventral hippocampal projection neurons. a** Schematic of dual-vector CRISPR tool to silence *FosB* in vHPC-NAc afferent neurons. Retrograde vector expressing Cre-dependent Cas9 (Cas9; red) is injected into NAc while local vector expressing Cre and *FosB* guide RNA (gRNA; green) is injected into vHPC. *FosB* silencing (*FosB* KO) occurs only in co-transduced vHPC-NAc neurons. **b** *FosB* silencing in vHPC-NAc neurons heightens stress-induced social avoidance. **P = 0.0092 (n = 6 control, n = 9 FosB KO; independent samples t-test compared to control). **c** *FosB* silencing in vHPC-NAc does not affect crossover latency in the passive avoidance learning paradigm (n = 7 control, n = 8 FosB KO), or **d** anxiety-like behavior in the elevated plus maze (EPM). **e** Schematic of dual-vector CRISPR silencing *FosB* in vHPC-BLA neurons. **f** *FosB* silencing in vHPC-BLA does not affect stress-induced social avoidance (n = 11 control, n = 10 FosB KO). **g** *FosB* silencing in vHPC-BLA impairs avoidance learning. #P = 0.0614 (n = 6 control, n = 8 FosB KO; two-way mixed ANOVA with Holm–Sidak post-tests compared to control). **h** *FosB* silencing in vHPC-BLA decreases anxiety-like behavior in the EPM, increasing time spent and entries into the open arms. **P = 0.0079 for time spent and **P = 0.0022 for entries (n = 5 control, n = 8 FosB KO; independent samples t-test compared to control). All graphs are represented as mean ± SEM.

vHPC-NAc afferents appears to be critical for a normal response to stress, and in its absence, mice are sensitized specifically to the social aspects of this stress response.

vHPC sends projections to other brain regions implicated in stress responses, including BLA, an area important in fear learning and anxiety[29–31]. In order to determine the circuit-specificity of our ΔFosB manipulations, we also assessed stress-induced social avoidance after *FosB* KO in the vHPC-BLA circuit (Fig. 2e). *FosB* KO in vHPC-BLA did not impair stress-induced social avoidance (Fig. 2f). Interestingly, however, it significantly impaired passive avoidance learning (Fig. 2g) and reduced anxiety-like behavior (Fig. 2h), with no change in locomotor activity (Fig. S6b). Similar observations, e.g. decreased anxiety, fear expression, and fear extinction, have been shown after vHPC lesions[2,31,32], which suggests that ΔFosB in vHPC-BLA projections is necessary for the expression of fear and anxiety and that

knockout of ΔFosB in this circuit is anxiolytic. While vHPC projections to NAc do send collaterals to other regions[3], we have found that these rarely overlap and the majority of NAc and BLA projections are separate populations (Fig. S7). Taken together, these data reveal that vHPC *FosB* gene products regulate phenotypes in a circuit-specific manner and provide rationale for previously dissonant findings about vHPC's role in stress behaviors.

The *FosB* gene encodes at least 3 isoforms, including full-length FosB, ΔFosB, and Δ2ΔFosB, and neuronal activity induces all of these isoforms[12,33–35]. It is therefore possible that any or all of these isoforms are important for the functional effects of *FosB* silencing in vHPC-NAc projections on stress-induced social avoidance. To investigate the specific role of ΔFosB in vHPC projections to NAc in stress-induced behaviors, we developed an additional intersecting viral strategy that would allow us to

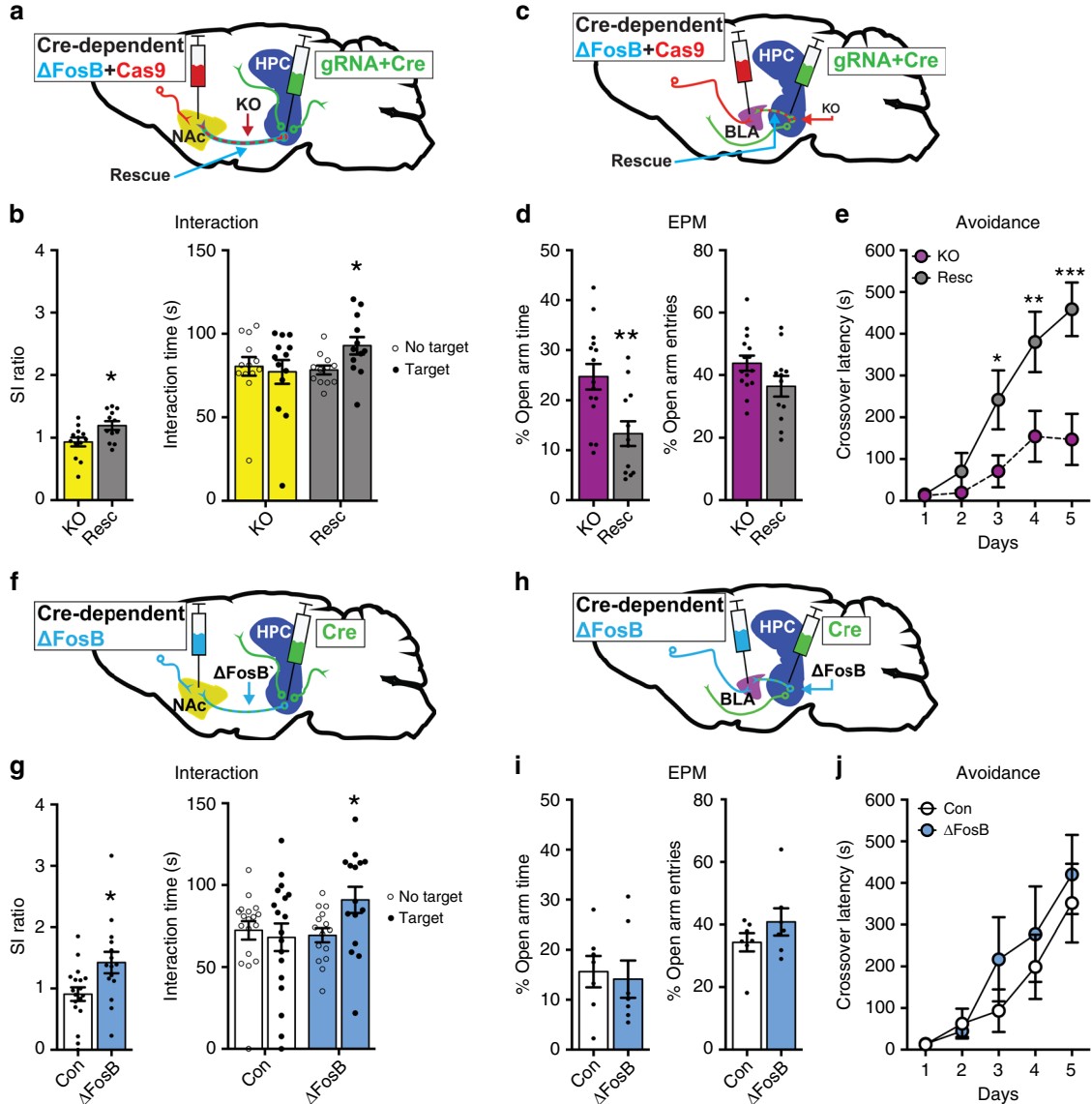

**Fig. 3 Circuit-specific vHPC ΔFosB mediates stress resilience and anxiety-like behavior. a** Schematic of ΔFosB rescue experiments in vHPC-NAc afferent neurons. Retrograde vector expressing Cre-dependent Cas9 and ΔFosB is injected into NAc while local vector expressing Cre and *FosB* guide gRNA is injected into vHPC. *FosB* silencing (KO) occurs in co-transduced vHPC-NAc neurons, and ΔFosB overexpression is also driven by Cre in the same neurons (Rescue). **b** ΔFosB rescue (Resc) in vHPC-NAc neurons reverses stress-induced social avoidance, indicated by enhanced social interaction ratio (left) and increased investigation of a social target (right). *$P = 0.0161$ for SI ratio and *$P = 0.0431$ for interaction time ($n = 13$ KO, $n = 12$ Resc; SI ratio: independent samples *t*-test compared to KO; investigation time: two-way mixed ANOVA with Holm–Sidak post-test KO No Target vs Target). **c** Schematic of ΔFosB rescue experiments in vHPC-BLA afferent neurons. **d** ΔFosB rescue in vHPC-BLA neurons reverses the anxiolytic effects of *FosB* silencing, decreasing open arm time in the EPM. **$P = 0.0042$ ($n = 15$ KO, $n = 12$ Resc; independent samples *t*-test compared to KO). **e** ΔFosB rescue also reverses avoidance learning impairment. *$P = 0.0459$, **$P = 0.0060$, ***$P < 0.0001$ ($n = 15$ KO, $n = 13$ Resc; two-way mixed ANOVA with Holm–Sidak post-tests compared to KO). **f** Schematic of ΔFosB overexpression in vHPC-NAc afferent neurons. Retrograde vector expressing Cre-dependent ΔFosB is injected into NAc while local vector expressing Cre is injected into vHPC. ΔFosB overexpression is driven by Cre only in co-transduced vHPC-NAc neurons (ΔFosB). **g** ΔFosB overexpression in vHPC-NAc enhances social interaction following stress, producing a resilient phenotype. *$P < 0.05$ ($n = 18$ control, $n = 12$ ΔFosB; SI ratio: independent samples *t*-test compared to control; interaction time: two-way ANOVA followed by Holm–Sidak post-test ΔFosB No Target vs Target). **h** Schematic of ΔFosB overexpression in vHPC-BLA afferent neurons. **i, j** ΔFosB overexpression in vHPC-BLA does not alter anxiety-like behavior or avoidance learning ($n = 7$ mice/group). All graphs are represented as mean ± SEM.

overexpress recombinant ΔFosB back into the same circuit in which we use CRISPR to silence *FosB* gene expression (Fig. 3a). These tools cause retrograde Cre-dependent ΔFosB and Cas9 expression (Resc) or Cre-dependent Cas9 expression alone (KO). We injected these into NAc, and subsequently injected local HSV expressing Cre and the FosB gRNA into vHPC of adult mice (Fig. 3a). Critically, ΔFosB rescue in *FosB* KO vHPC-NAc neurons reversed the enhancement in stress-induced social avoidance

(Fig. 3b). Similarly, ΔFosB rescue in *FosB* KO vHPC-BLA neurons (Fig. 3c) blocked the *FosB* KO-mediated reduction in basal anxiety (Fig. 3d) and impairment in avoidance learning (Fig. 3e). To control for the presence of CRISPR-Cas9-mediated *FosB* KO, ΔFosB was independently overexpressed in these same circuits by injecting retrograde Cre-inducible ΔFosB in NAc or BLA (HSV-hEF1α-LSIL-ΔFosB) and persistent Cre (AAV2-CMV-Cre-GFP) into vHPC (Fig. 3f, h), which produces ΔFosB overexpression

specifically in vHPC-NAc-projecting cells (Fig. S8). In the absence of *FosB* KO, ΔFosB overexpression did not affect basal anxiety (Figs. 3i, S9a) or avoidance learning (Figs. 3j, S9b) in either projection. Interestingly, we found that overexpression of ΔFosB in vHPC-NAc produced a resilient phenotype to stress-induced social avoidance (Fig. 3g). Together these data elucidate a clear role for ΔFosB in vHPC projections: ΔFosB in vHPC-NAc is necessary and sufficient for resilience to stress-induced social avoidance, and ΔFosB in vHPC-BLA is necessary for the expression of fear and anxiety. Moreover, the complete behavioral rescue by ΔFosB overexpression in the same circuit in which we use the novel CRISPR system to silence the *FosB* gene indicates that our behavioral effects are indeed mediated by *FosB* knockout, and not through off-target effects of the CRISPR system. Our results demonstrate that individual genes can have disparate roles within a single brain structure based not only on heterogenous cell types but on the specific projections of the neurons in which they are expressed, which is a significant step toward illuminating the role of vHPC circuits in stress-related neuropsychiatric disease and a key finding to expand our understanding of the molecular mechanisms of complex brain functions and behaviors.

As previous findings demonstrated that increased activity in vHPC-NAc neurons underlies CSDS susceptibility[4], we next sought to ascertain the role of ΔFosB in the physiological properties of the vHPC-NAc circuit. We first examined the effects of ΔFosB overexpression in putative glutamatergic pyramidal neurons of vCA1 via whole-cell, patch-clamp electrophysiology in ex vivo slice preparations[17,18]. Viral-mediated ΔFosB overexpression reduced the excitability of vCA1 neurons, indicated by a decrease in the number of current-elicited spikes across increasing step currents (Fig. 4a, b) and an increase in rheobase (Fig. 4c). Little or no change was observed in most cellular properties (Table S1), but ΔFosB overexpression decreased outward rectification of the $I$–$V$ curve (Fig. S10a, b). Moreover, ΔFosB decreased amplitude and increased half-width of evoked action potentials and decreased the frequency of spontaneous synaptic currents (Fig. S10c–f), similar to what we have previously observed in dHPC CA1 neurons[18]. This suite of observations suggests that ΔFosB expression reduces the excitability of vHPC neurons, likely via changes in intrinsic membrane properties. Importantly, in the vHPC-NAc circuit, stress-induced increases in ΔFosB could cause the decreased excitability previously shown to drive resilience to CSDS[4], thus providing a novel molecular mechanism for this important circuit-level underpinning of stress resilience.

In order to determine whether ΔFosB regulates excitability of vHPC-NAc projection neurons specifically, we crossed *floxed FosB* mice (*FosB$^{fl/fl}$*)[36] with the GFP-L10a line to generate floxed FosB/GFP-L10a mice (*FosB* KO) and non-floxed GFP-L10a littermate controls (WT). We used the same described viral retrograde Cre strategy to drive GFP-L10a expression in vHPC-NAc neurons of all mice and knock out ΔFosB expression in vHPC-NAc neurons of the KO, but not WT, mice (Fig. 4d–f and Fig. S11). This method is not circuit-specific (see Suppl Fig. S11c) thus making it unsuitable for assessment of stress-induced social avoidance (Suppl. Fig. S11d), yet it allows us to determine functional effects of ΔFosB KO on vHPC-NAc. Whole-cell ex vivo slice recordings from GFP-expressing vCA1-NAc neurons showed that *FosB* KO produces hyperexcitability in NAc-projecting vHPC neurons (Fig. 4g–i). *FosB* KO increased outward rectification of the $I$–$V$ curve, increased amplitude and decreased half-width of evoked action potentials and increased the frequency but decreased amplitude of spontaneous synaptic currents (Fig. S12), essentially creating the opposite phenotype from ΔFosB overexpression (Figs. 4a–c and S10). As reduced activity in vHPC glutamatergic neurons synapsing onto NAc

MSNs confers resilience to CSDS and increased activity enhances susceptibility[4], and our results above show that vHPC-NAc ΔFosB is necessary for CSDS resilience, this work supports a model in which stress-induced ΔFosB promotes hypoexcitability of NAc-projecting vHPC neurons to drive resilience to CSDS-induced social avoidance.

In order to uncover potential mechanisms by which ΔFosB regulates vHPC-NAc excitability and related behaviors, we interrogated gene expression regulated by ΔFosB in NAc-projecting vHPC neurons using our circuit-specific TRAP strategy in WT and *FosB* KO L10-GFP mice (Fig. 5a). Purified vHPC-NAC specific mRNA was used to prepare cDNA libraries that were sequenced. We found that there was significant enrichment of neuron-specific mRNA (like *kalrn*) in our TRAP samples compared to input controls, with no change in enrichment of genes common to all cells (like actin; Fig. S13), indicating that our TRAP technique successfully isolated mRNA from neurons. Compared to mRNA purified from WT controls, mRNA from *floxed FosB* mice showed a nearly complete absence of *FosB* gene expression (Fig. 5b), indicating that our purified mRNA comes from vHPC neurons projecting to NAc, as retrograde viral infection in NAc is the only source of Cre in these animals. When we compared mRNAs differentially enriched in WT vs *floxed FosB* vHPC-NAc neurons, we uncovered hundreds of potential ΔFosB gene targets (Tables S2 and S3). In order to begin validation of some of the most interesting potential targets, we overexpressed ΔFosB in Neuro2a cells and used qPCR to assess expression of target genes (Fig. 5c). ΔFosB overexpression in cell culture regulated a number of genes identified in the vHPC-NAc TRAP-Seq, including *Nefm* and *Prkcb*. We chose to examine *Adra2a*, which encodes the α2a adrenergic receptor (α2AAR), a gene upregulated in the vHPC-NAc TRAP (Fig. S13) to highlight our approach to possible interrogation of downstream targets. Norepinephrine is associated with stress-related diseases such as depression[37], as are α2AARs[38,39], and α2AAR density is increased in the hippocampus of depressed patients[38,40] and mediates the effects of antidepressants[41,42]. Critically, α2AARs enhance the excitability of pyramidal neurons[43]. Therefore, it provided a potentially viable target that may underlie ΔFosB's effects in this circuit. We found that *Adra2a* mRNA is downregulated in Neuro2a cells after ΔFosB overexpression (Fig. 5c), indicating that ΔFosB represses the *Adra2a* gene, as it represses other genes in dHPC[44,45]. In order to confirm this regulation, Neuro2a cells were transfected with Cas9 and FosB gRNA or the ΔFosB transcriptional repressor ΔJunD, and both conditions caused an increase in *Adra2a* mRNA (Fig. 5d), confirming that ΔFosB represses this gene in cultured cells. To validate this observation in vHPC-NAc neurons in vivo, we injected retrograde Cre in NAc of our GFP reporter animals and immunostained for α2AAR and GFP in both WT and *floxed FosB* vHPC-NAc neurons. We found that *FosB* KO increased the expression of α2AAR in vHPC-NAc projections (Fig. 5e, f), confirming that ΔFosB directly downregulates *Adra2a* in NAc-projecting vHPC neurons. Thus ΔFosB-dependent repression of α2AAR expression in vHPC-NAc could potentially be a mechanism underlying ΔFosB effects. However, considering ΔFosB also regulates a variety of gene targets, further investigation into *Adra2a* and many other genes is warranted.

## Discussion

The current findings indicate that ΔFosB is induced in response to stress in glutamatergic vHPC neurons projecting to NAc, altering gene expression to reduce circuit excitability and drive resilience to stress-induced social avoidance (Fig. 5g). As we observed that ΔFosB alters synaptic inputs onto vHPC-NAc neurons, it is also possible that ΔFosB may mediate

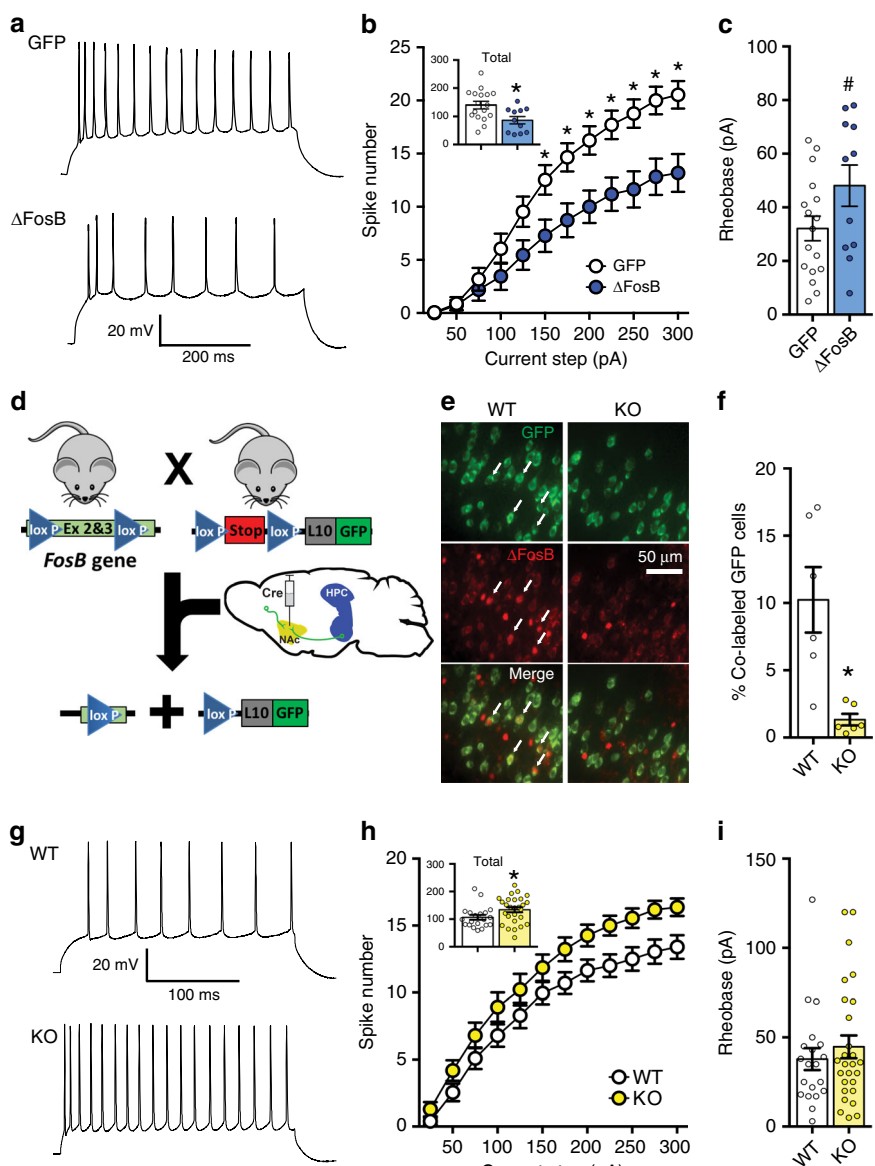

**Fig. 4 ΔFosB regulates the cellular excitability of NAc-projecting vHPC neurons. a** Representative voltage traces from 200 pA depolarizing current injection in ventral hippocampus CA1 (vCA1) neurons expressing GFP or GFP+ΔFosB (ΔFosB). **b** ΔFosB reduces the number of elicited spikes in vCA1 neurons across increasing current steps (25–300 pA) or the total spikes for all steps (inset). *$P = 0.0108$ for inset; *$P = 0.0345$ for 150 pA, *$P = 0.0135$ for 175 pA, *$P = 0.0088$ for 200 pA, *$P = 0.0058$ for 225 pA, *$P = 0.0020$ for 250–275 pA, and *$P = 0.0015$ for 300 pA ($n = 17$ GFP cells, $n = 11$ ΔFosB cells; total: independent samples $t$-test; across steps: two-way mixed ANOVA with Holm–Sidak post-tests compared to GFP). **c** ΔFosB trends toward increasing rheobase. #$P = 0.0680$ (independent samples $t$-test compared to GFP). **d** Schematic of FosB knockout in NAc-projecting neurons. Floxed FosB mice ($FosB^{fl/fl}$; removal of exon 2 & 3) were crossed with mice expressing Cre-dependent GFP-L10 ribosomal fusion protein ($Rosa26^{eGFP-L10a}$) to generate floxed FosB/GFP-L10 mice (KO) and non-floxed GFP-L10 littermate controls (WT). Retrograde Cre vector was injected in NAc to both drive GFP-L10 expression and knock out FosB in NAc-projecting neurons. **e** Representative coronal vHPC images (×20) showing NAc-projecting neurons expressing GFP (green, top), ΔFosB (red, middle), and merge (bottom). White arrows indicate co-labeled GFP+ ΔFosB-expressing cells. WT (left) expressed more ΔFosB co-labeled GFP projections compared to KO quantified in (**f**), KO significantly reduces the % of co-labeling of GFP+ ΔFosB-expressing vHPC-NAc neurons. *$P = 0.0048$ ($n = 6$ mice/group; independent samples t-test compared to WT). **g** Representative voltage traces from 200 pA depolarizing current injection in NAc-projecting GFP-labeled vCA1 neurons from WT and KO mice. **h** KO of FosB increases the number of elicited spikes in NAc-projecting vCA1 neurons across increasing current steps (25–300 pA) or the total spikes for all steps (inset). *$P = 0.0418$ for inset ($n = 20$ WT cells, $n = 27$ KO cells; total: independent samples $t$-test; across steps: two-way mixed ANOVA with Holm–Sidak post-tests versus WT). **i** No difference in rheobase was observed between WT and KO neurons ($n = 20$ WT cells, $n = 27$ KO cells). All graphs are represented as mean ± SEM.

stress-associated synaptic plasticity in vHPC. In keeping with this possibility, CSDS, like ΔFosB, increases the number of thin immature dendritic spines on dorsal hippocampal CA1 neurons[17,46], and decreases AMPA receptor subunit levels at the postsynaptic density[46]. Indeed, many postsynaptic structural/signaling proteins were uncovered in our screen of vHPC-NAc

ΔFosB gene targets (Tables S2 and S3), and future studies will use these data to explore the mechanisms by which stress alters both intrinsic excitability and synaptic inputs onto the vHPC-NAc and vHPC-BLA circuits. Ventral hippocampal circuits are key regulators of a variety of emotionally driven behaviors and hold promise for future treatments of neuropsychiatric disease if we

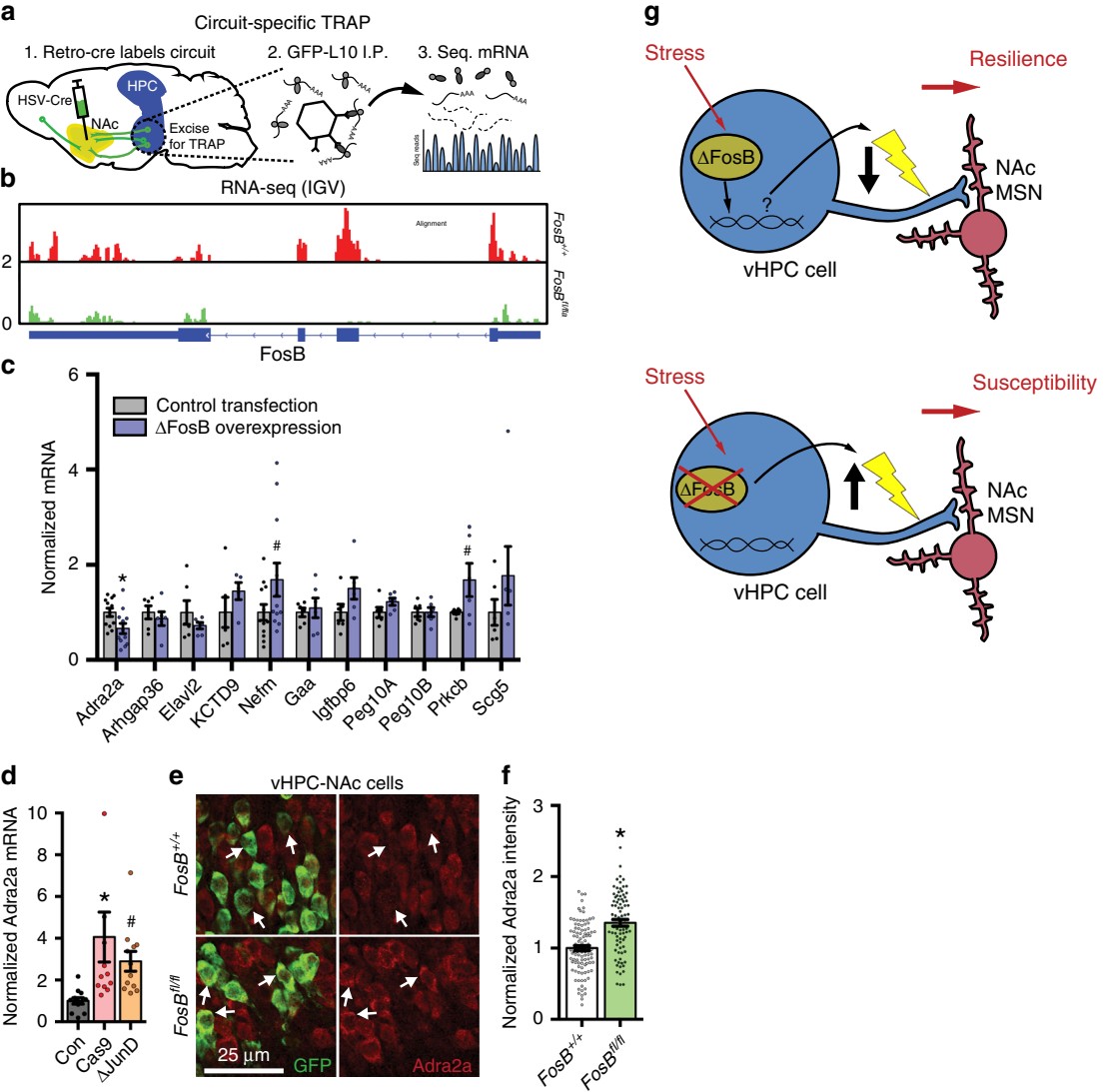

**Fig. 5 ΔFosB orchestrates gene expression in NAc-projecting vHPC neurons. a** Experimental design for circuit-specific TRAP. Retrograde Cre vector is injected into NAc and vHPC is harvested for immunoprecipitation of ribosomes from L10-GFP-expressing NAc-projecting cells and sequencing of actively translating mRNA. **b** Sequencing reads (red or green peaks) for *FosB* gene (exons indicated by thick blue bars beneath graph) from WT (FosB$^{+/+}$) and *FosB* KO (FosB$^{fl/fl}$) vHPC-NAc neuron mRNA. **c** ΔFosB overexpression in Neuro2A cells regulates mRNA expression of some TRAP-identified target genes. *$P = 0.0240$ for *Adra2a*, #$P = 0.0914$ for *Nefm*, #$P = 0.0815$ for *Prkcb* (independent samples *t*-test compared to Control Transfection; *Adra2a* $n = 12$ plates/ group; *Arhgap36* $n = 6$; *Elavl2* $n = 6$; *Kctd9* $n = 5-6$; *Nefm* $n = 12$; *Gaa* $n = 6$; *Igfbp6* $n = 6$; *Peg10a* $n = 6$; *Peg10b* $n = 6$; *Prkcb* $n = 6$; *Scg5* $n = 6$). **d** *Adra2a* mRNA expression is increased in Neuro2A cells transfected with ΔJunD or *FosB*-specific gRNA and Cas9 (Cas9). *$P = 0.0133$, #$P = 0.0822$ ($n = 12$ wells/ group; one-way ANOVA with Holm–Sidak post-tests versus control). **e** Representative coronal vHPC images (20X) showing NAc-projecting neurons expressing GFP (green) and α2AAR (red) in *floxed FosB* KO (FosB$^{fl/fl}$; bottom panels) and WT (FosB$^{+/+}$; top panels) mice. **f** Intensity of α2AAR expression in NAc-projecting GFP-labeled is decreased by *FosB* KO (FosB$^{fl/fl}$). *$P < 0.0001$ ($n = 100$ FosB$^{+/+}$ cells, $n = 83$ FosB$^{fl/fl}$ cells; independent samples *t*-test compared to FosB$^{+/+}$). **e** Proposed model: in resilient mice, stress strongly induces ΔFosB in vHPC-NAc neurons initiating gene expression that underlies a protective decrease in excitability of the projection; in susceptible mice, a lack of *FosB* expression allows stress-associated differential gene expression to keep vHPC-NAc excitability high, driving social avoidance. All graphs are represented as mean ± SEM.

can reveal the molecular mechanisms of their regulation. In keeping with this, recent clinical studies have demonstrated that coherence of the hippocampal-amygdala circuit correlates with anxiety phenotypes in patients[47]. However, the mechanisms for differences in circuit coherence cannot yet be explored in humans, emphasizing the need for preclinical models and circuit-based tools for uncovering molecular mechanisms that could yield potential treatments. Although it is possible that drugs targeting ΔFosB could be developed, we have adopted the strategy that ΔFosB acts as an orchestrator of disease-relevant changes in gene expression, and thus uncovering the downstream targets of

ΔFosB transcriptional regulation may reveal myriad druggable pathways for the systemic or circuit-specific treatment of depression, potentially for symptoms of social withdrawal.

## Methods
**Animals**. All experiments were approved by the Institutional Animal Care and Use Committee at Michigan State University in accordance with AAALAC. Male C57Bl/6J mice (3–5/cage, 7–8-week old upon arrival from Jackson Labs) were allowed at least 5 d to acclimate to the facility prior to any experimental procedures. The *floxed FosB* mouse strain (FosB$^{fl/fl}$)[36] was a generous gift from the laboratory of Dr. Eric Nestler at the Icahn School of Medicine at Mount Sinai, and the Rosa26$^{eGFP-L10a}$ mice[48] were a generous gift from the laboratory of Dr. Gina

Leinninger at Michigan State University. Male CD1 retired breeder mice (1/cage; age varies upon arrival from Charles River) were allowed at least 1 week, but more commonly 2–3 weeks, to acclimate to the facility prior to any experimental procedures. Prior to CSDS experiments, CD1 was allowed 5–7 d to acclimate to the novel housing conditions described below. Unless otherwise stated, all mice were group housed in a 12:12 h light/dark cycle with ad libitum food and water. Temperature (22 °C) and humidity (50–55%) were held constant in animal housing and behavioral testing rooms.

**Surgery and viral vectors**. Stereotaxic surgery was conducted as previously described[17]. For ΔJunD experiments, viral vectors (AAV2-CMV-GFP or AAV2-CMV-ΔJunD-GFP) were bilaterally infused at two sites in the dHPC (−2.2 AP, ±2.0 ML, −2.1 and −1.9 DV relative to bregma, 10° angle) or vHPC (−3.6 AP, ±3.2 ML, −4.8 and −3.0 DV relative to bregma, 5° angle; 0.3 µL per DV site) to ensure spread of transduction throughout each structure. Experimental procedures commenced at least 4 weeks following surgery. For dual virus CRISPR/Cas9 *FosB* gene silencing experiments, Cas9-expressing retrograde vector (HSV-hEf1α-LS1L-myc-Cas9; 0.5 µL) was infused into NAc (+1.6 AP, ±1.5 ML, −4.4 DV relative to bregma, 10° angle) or BLA (−1.6 AP, ±3.4 ML, −4.5 DV relative to bregma, 0° angle). After 3 weeks, control viral vector (HSV-IE4/5-TB-eYFP-CMV-IRES-Cre) or *FosB* gRNA (HSV-IE4/5-TB-FosB gRNA-CMV-eYFP-IRES-CRE) were infused into the ventral CA1 region of vHPC (vCA1; −3.4 AP, ±3.2 ML, −4.8 DV relative to bregma, 3° angle; 0.5 µL). Experimental procedures commenced at least 2 weeks following vHPC surgeries. For rescue experiment, Cre-dependent Cas9 and ΔFosB-expressing vector (HSV-hEF1α-LSIL-ΔFosB-myc-Cas9; or Cre-dependent Cas9 alone vector as a control) was infused into NAc (0.5 µL), and following 3 weeks *FosB* gRNA vector was infused into vCA1 (0.5 µL). For ΔFosB overexpression in hippocampal circuits, Cre-dependent ΔFosB vector (HSV-hEF1α-LSIL-ΔFosB-IRES-GFP; 0.5 µL) was infused into NAc or BLA and Cre vector (AAV2-CMV-Cre-GFP; or AAV2-CMV GFP as a control) was infused into vCA1. Experimental procedures commenced at least 4 weeks following surgeries. For electrophysiology of ΔFosB-expressing vHPC neurons, viral vectors (HSV-IE4/5-ΔFosB-CMV-GFP; or HSV-CMV-GFP as control) were bilateral infused into vCA1 (0.5 µL) and experimental procedures commenced 2–4 d following surgery. For electrophysiology of *FosB* KO in vHPC-NAc neurons, Cre expressing retrograde vector (HSV-hEf1α-Cre) was bilaterally infused into NAc (0.5 µL) of *Rosa26*[eGFP/L10] mice and experimental procedures commenced at least 3 weeks following surgery. AAV viral vectors were obtained from the University of North Carolina at Chapel Hill (UNC Vector Core: https://www.med.unc.edu/genetherapy/vectorcore/) and HSV viral vectors came from Massachusetts General Hospital (Dr. Rachael Neve, Gene Delivery Technology Core: https://researchcores.partners.org/mvvc/about).

**Chronic social defeat stress**. CSDS was performed as previously described[21,24–26]. In brief, mice were placed into the homecage of an aggressive retired breeder CD1 mouse containing a perforated plexiglass divider placed between the walls of the cage. The experimental mice were allowed to physically interact with the CD1 for 10 min Following the aggressive encounter, the mice were placed into the other side of the divider from the CD1 aggressor mouse allowing sensory, but not physical, contact for 24 h. This protocol was repeated daily for 10 d with a new aggressor every day. Behavioral testing began the day following the final day of stress.

**Subchronic defeat stress**. Also called a microdefeat, subchronic defeat is an abbreviated social defeat stress protocol[24,25]. Mice were placed in the homecage of an aggressive retired breeder CD1 mouse and allowed to interact for 3–5 min, then removed and allowed to rest in their homecage for 15 min This was repeated for a total of 3 consecutive encounters in a single day. Behavioral testing began the following day.

**Behavioral testing**. Behavior was collected using a IR-CCD camera (Panasonic) and analyzed using automated videotracking software (CleverSys). Animals were transported to the behavioral testing rooms in their home cages and allowed 30 min to habituate to the room.

**Social interaction**. SI testing was conducted as previously described[24,26]. Briefly, under red light conditions, mice were placed into the center of a custom-made square, opaque arena (38 cm W × 38 cm L × 35 cm H) containing an empty wire mesh cage (10-cm diameter) against one wall and allowed to explore for 150 s. The experimental mice were then removed from the arena and a novel CD1 mouse was placed in the wire mesh cage. Experimental mice were then reintroduced to the arena and allowed to explore for another 150 s. The time spent in proximity (7.5 cm) of the wire mesh object was defined as "interaction zone" time while the time spent in two corners (9 × 9 cm square) farthest from the object were defined as corner zone time. SI ratio was determined by calculating the time spent in the interaction zone when the CD1 was present divided by the time spent in the interaction zone when the CD1 was absent.

**Elevated plus maze (EPM)**. Testing in EPM was conducted as previously described[17]. Briefly, under red light conditions, mice were placed onto the center of an EPM, with two open arms and two closed arms, and allowed to explore for 5 min. The time spent on the open arms and the number of open arm entries were recorded.

**Temporally dissociative passive avoidance (TDPA)**. TDPA testing was conducted as previously described[17,49]. Briefly, mice were placed into the lit side of light dark box. After 2 min of exploration, a door allowing entry into the dark side was raised. Upon entry (full body, excluding tail) into the dark side, the door was lowered and, after 5 min, mice received a mild footshock (0.7 mA, 2 s). Mice were returned to their homecage after 30 s. Testing (all of the same conditions including footshock) was repeated daily for 5 d. The latency (s) to "cross over" from the light side to dark side was manually recorded daily.

**Open field (OF)**. Testing in OF was conducted as previously described[17]. Briefly, under red light conditions, mice were placed into the center of white opaque, square custom-made OF and allowed to explore for 1 h. Time spent in the center zone (50% of the size of OF, centered) and the total distance moved (in cm) were recorded.

**von Frey test for tactile allodynia**. Tactile allodynia was assessed using calibrated von Frey filaments as previously described[50]. Mice were habituated to clear plexiglass containers over a mesh floor for 1 h. The next day mice were placed back into the containers for 30 min prior to testing. A series of von Frey filaments were applied to the plantar surface of the hindpaw with sufficient force to bend the filaments for at least 6 s. A paw withdrawal or rapid flinch response was recorded. In the absence of this response, a filament of the next greater force was applied. If a response occurred, the next lower filament was applied. Paw withdrawal threshold (g) was recorded as the force that produced a 50% likelihood of withdrawal.

**Novelty-suppressed feeding**. Mice were restricted from ad libitum chow for 24 h prior to testing. Mice were placed into the corner of an OF arena (see above) with one specific difference: a single chow pellet was placed into the center under light conditions. All mice were naive to the arena. Latency to feed (s) was visually recorded as a measure of novelty-induced suppression of feeding.

**Single-trial contextual avoidance**. Contextual avoidance after a single conditioning trial was assessed under the same conditions as TDPA with one exception: mice were shocked immediately (1 s) after crossover into the dark side of the box. Crossover latency was manually recorded 24 h later.

**Tail suspension test**. Mice were hung using standard laboratory tape from their tails. Tape was adhered to a horizontal bar. Mice were allowed to hang for 10 min Any mouse that crawled back up its tail was removed from the analysis. Immobility, defined as lack of skeletal movement for at least 1 s, was recorded for the duration of the 10 min via automated tracking software (FreezeScan, CleverSys, Inc.)

**Immunofluorescent staining for FosB-immunoreactivity**. *Rosa26*[eGFP/L10a] mice underwent stereotaxic surgery to infuse retrograde HSV-hEf1α-Cre into the NAc (see above). After waiting three weeks for full expression, mice were transcardially perfused with cold PBS, followed by 10% formalin. In other experiments, mice received ΔFosB overexpression (see above) and were sacrificed and perfused 4–8 weeks following surgery. Brains from all immunostaining experiments were postfixed 24 h in 10% formalin, cryopreserved in 30% sucrose, and sliced frozen on an SM2010R microtome (Leica) into 35-µm sections. Immunohistochemistry was performed using primary antibodies against FosB (ab11959; 1:1000; Abcam), GFP (ab5450; 1:1000; Abcam), and α2AAR (PA1-048; 1:1000; Invitrogen), and secondary antibodies (1:200; Jackson Immunoresearch) conjugated to fluorescent markers (AlexaFluor 488; Cy3; Cy5). Fluorescent images were visualized on an Olympus FluoView 1000 filter-based laser scanning confocal microscope. Intensity of α2AAR signal in individual cells was quantified using ImageJ software by an experimenter blinded to conditions.

**Translational ribosomal affinity purification (TRAP) and cDNA library preparation**. Three weeks following injection of retrograde HSV-Cre into NAc, Cre-dependent L10-GFP-expressing mice (*Rosa26*[eGFP/L10a]) were sacrificed and brains were immediately dissected into 1-mm coronal sections. Transduced tissue from ventral hippocampi (vHPC) of both wild-type and *FosB*[fl/fl] mice was collected using 14-gauge biopsy punches guided by a fluorescent dissecting microscope (Leica) and stored at −80 °C until processing (n = 3/group, 3–4 mice pooled per n). Polyribosome-associated RNA was affinity purified as previously described[51,52]. Briefly, tissue was homogenized in ice-cold tissue-lysis buffer (20 mM HEPES [pH 7.4], 150 mM KCl, 10 mM MgCl₂, 0.5 mM dithiothreitol, 100 µg/ml cycloheximide, protease inhibitors, and recombinant RNase inhibitors) using a motor-driven Teflon glass homogenizer. Homogenates were centrifuged for 10 min at 2000 × g (4 °C), supernatant was supplemented with 1% NP-40 (AG Scientific, #P1505) and 30 mM DHPC (Avanti Polar Lipids, #850306P), and centrifuged again for 10 min at 20,000 × g (4 °C). Supernatant was collected and incubated with Streptavidin

MyOne T1 Dynabeads (Invitrogen, #65601) that were coated with anti-GFP antibodies (Memorial Sloan-Kettering Monoclonal Antibody Facility; clone names: Htz-GFP-19F7 and Htz-GFP-19C8, 50 μg per antibody per sample) using recombinant biotinylated Protein L (Thermo Fisher Scientific, #29997) for 16–18 h on a rotator (4 °C) in low salt buffer (20 mM HEPES [pH 7.4], 350 mM KCl, 1% NP-40, 0.5 mM dithiothreitol, 100 μg/ml cycloheximide). Beads were isolated and washed with high salt buffer (20 mM HEPES [pH 7.4], 350 mM KCl, 1% NP-40, 0.5 mM dithiothreitol, 100 μg/ml cycloheximide) and RNA was purified using the RNeasy MicroKit (Qiagen, #74004). In order to increase yield, each RNA sample was initially passed through the Qiagen MinElute™ column three times. Following purification, RNA was quantified using a Qubit fluorometer (Invitrogen) and RNA quality was analyzed using a 4200 Agilent Tapestation (Agilent Technologies). cDNA libraries from 5 ng total RNA were prepared using the SMARTer® Stranded Total RNA-Seq Kit (Takara Bio USA, #635005), according to manufacturer's instructions. cDNA libraries were pooled following Qubit measurement and TapeStation analysis, with a final concentration ~7 nM.

**Sequencing**. Sequencing was performed at the Icahn School of Medicine at Mount Sinai Genomics Core Facility (https://icahn.mssm.edu/research/genomics/core-facility). Raw sequencing reads from mice were mapped to mm9 using TopHat[53]. Counts of reads mapping to genes were obtained using HTSseq-counts software[54] against Gencode vM1 (mm9) annotation. Differential expression was done using the DESeq2 package[55].

**Electrophysiology**. Whole-cell, ex vivo slice electrophysiology was conducted as previously described[18]. All solutions were bubbled with 95% $O_2$–5% $CO_2$ throughout the procedure. Mice were anesthetized with isoflurane anesthesia and transcardially perfused with sucrose artificial cerebrospinal fluid (aCSF; in mM: 234 sucrose, 2.5 KCl, 1.25 $NaH_2PO_4$, 10 $MgSO_4$, 0.5 CaCl, 26 $NaHCO_3$, 11 glucose). Brains were rapidly removed, blocked, and placed in cold sucrose aCSF. Coronal sections (250 μM) containing vHPC were cut on a vibratome (Leica) and transferred to an incubation chamber containing aCSF (in mM: 126 NaCl, 2.5 KCl, 1.25 $NaH_2PO_4$, 2 $MgCl_2$, 2 $CaCl_2$, 26 $NaHCO_3$, 10 glucose) held at 34 °C for 30 min before moving to aCSF at room temperature until used for recordings. Recordings were made from a submersion chamber perfused with aCSF (2 mL/min) held at 32 °C. Borosilicate glass electrodes (3–6 MΩ) were filled with K-gluconate internal solution (in mM: 115 potassium gluconate, 20 KCl, 1.5 MgCl, 10 phosphocreatine-Tris, 2 MgATP, 0.5 $Na_3GTP$; pH 7.2–7.4; 280–285 mOsm). GFP-positive cells in the ventral CA1 region of HPC were visualized using an upright microscope (Olympus) using infrared and epifluorescent illumination. Whole-cell patch-clamp recordings were made from transfected cells using a Multiclamp 700B amplifier and Digidata 1440A digitizer (Molecular Devices) and whole-cell junction potential was not corrected. Traces were sampled (10 kHz), filtered (10 kHz), and digitally stored. Cells with membrane potential more positive than −50 mV or series resistance >20 MΩ were omitted from analysis. Rheobase was measured by giving brief (250 ms) depolarizing (5 pA) steps with 250 ms between steps. Elicited spike number was measured by issuing increasing depolarizing steps (25–300 pA, 500 ms) with 500 ms step intervals. For synaptic recordings of spontaneous excitatory postsynaptic currents (sEPSCs), cells were held at −80 mV for 2 min All electrophysiology recordings were made at approximately 30–32 °C by warming the aCSF line with a single inline heater (Warner Instruments).

**Immunohistochemistry for detection of ΔFosB following chronic stress or fluoxetine treatment**. Immunohistochemistry was performed as previously described[17], and CSDS is described above. A separate cohort of mice received once-daily intraperitoneal injections of fluoxetine (20 mg/kg; dissolved in saline) or saline. For both experiments, mice were sacrificed 24 h following the last injection or defeat episode. Animals were transcardially perfused with cold PBS, followed by 10% formalin. Brains were postfixed 24 h in 10% formalin, cryopreserved in 30% sucrose, followed by slicing on a microtome into 35-μm sections. Immunohistochemistry was performed using anti-FosB primary antibody (2251; 1:500; Cell Signaling) and biotin-conjugated secondary anti-rabbit (BA-1000; 1:1000; Vector) then visualized by 3,3′-diaminobenzidine staining (Vector Laboratories).

**Detection of ventral hippocampal afferents to NAc**. mCherry expressing retrograde vector (HSV-hEf1α-mCherry) was infused into NAc and perfused coronal sections were taken at 3 weeks following surgery according to previously described protocol (see above).

**CRISPR Guide RNA design and testing**. gRNAs targeting exon 2 of the FosB gene were designed using e-CRISP software (www.e-CRISP.org). The top four sequences were:

gRNA1: TACACCGGGAGCCGGAGTCG
gRNA2: TTACGATCTAAAACTTACCT (this gRNA was most effective and was selected for all in vivo work described in the current manuscript; also referred to as AJR4 as it was the fourth gRNA produced for our lab)
gRNA3: TCAACATCCGCTAAGGAAGA
gRNA4: CCGTCTTCCTTAGCGGATGT

Each gRNA was tested by transfection in a mammalian expression plasmid also containing Cas9. Briefly, Neuro2a cells (N2a, American Type Culture Collection) were cultured in EMEM (ATCC) supplemented with 10% heat-inactivated fetal bovine serum (ATCC) in a 5% $CO_2$ humidified atmosphere at 37 °C. Cells were plated into 12-well plates, and 24 h later (when cells were ~30% confluent) cells were transiently transfected using Effectene (Qiagen) with a total of 200 ng DNA per well. Cells were transfected with empty vector, Cas9 alone, or Cas9 with a gRNA to be tested. Cells were then serum starved for 24 h, then refed for 4 h to induce FosB gene expression. Cells were pelleted, samples were run on gradient polyacrylamide gels and transferred to PVDF membranes, and Western blot was performed using rabbit anti-FosB antibody (2251; 1:500; Cell Signalling) and HRP conjugated anti-rabbit secondary (PI-1000; 1:40,000; Vector). Signal was detected on film and quantified using ImageJ software.

**T7 endonulease surveyor analysis**. T7 surveyor analysis was performed essentially as described[56,57]. Briefly, Neuro2a cells were plated and transfected as described above, and DNA was extracted using QuickExtract solution (Epicentre Biotechnologies). A region of the FosB gene containing the site targeted by our gRNA was amplified by PCR using the following primers:

RB936: GCTTTTCCCGGAGACTACGA
RB937: AAACCAAAGTGCAAACCGAAC

The surveyor nuclease was then used to selectively digest mismatched duplex formed from the PCR products, allowing detection of Cas9 mutated DNA.

**Single virus CRISPR/Cas9 FosB KO in brain**. Male C57Bl/6J mice (8–10 weeks) received surgeries in dHPC (see Methods for stereotaxic surgery above) infusing HSV expressing both Cas9 and FosB gRNA (HSV-syn-Cas9-gRNA-IRES-GFP; or HSV-CMV-GFP as a control). Mice were then tested 2 d following surgery for novel object recognition as previously described[17]. Briefly, mice were exposed for 30 min to two similar, familiar objects in an OF over 2 consecutive days. On the next day, a 5 min test for recognition was conducted, where a novel object was placed instead of one of the familiar objects. Time spent in a zone (interaction time) around the novel and familiar object was measured. Following testing, animals were sacrificed by cardiac perfusion and immunofluorescent detection of FosB and GFP expression was conducted (as above).

**Validation of dual virus CRISPR/Cas9 FosB KO in vHPC projections**. Male C57Bl/6J mice (8–10 weeks) received surgeries in NAc or BLA (see Methods for stereotaxic surgery above) infusing HSV expressing Cre-dependent Cas9 (see Methods for stereotaxic surgery above) and 3 weeks later local HSVs expressing Cre and FosB gRNA into vHPC. Four days post gRNA viral injections, mice were sacrificed according to methods for immunohistochemistry above. Coronal sections containing vHPC were stained for Cas9 (in red; 1:500 Diagenode C15200229) and FosB (in cyan; 1:500 ab11959). FosB intensity levels for each Cas9+ neuron were measured using ImageJ, and the numbers of FosB+ and FosB-Cas9-expressing neurons were quantified.

**Quantitative PCR from Neuro2A cells**. Neuro2A cells were treated and harvested according to prior method (see CRISPR Guide RNA design and testing), except cells were transfected with empty vector (control), ΔFosB, Cas9 + FosB gRNA, or ΔJunD. After cells were harvested, RNA was isolated using TriZol (Invitrogen) homogenization and chloroform layer separation. The clear RNA layer was then processed (RNAeasy MicroKit, Qiagen #74004) and analyzed with NanoDrop. A volume of 10 μL of RNA was reverse transcribed to cDNA (High Capactiy cDNA Reverse Transcription Kits Applied BioSystens #4368814). Prior to qPCR, cDNA was diluted to 200 μL. The reaction mixture consisted of 10 μL PowerSYBR Green PCR Master Mix (Applied Biosystems; #436759), 2 μL each of forward and reverse primers and water, and 4 μL cDNA template. Samples were then heated to 95 °C for 10 min (Step 1) followed by 40 cycles of 95 °C for 15 s, 60 °C for 15 s, and 72 °C for 15 s (Step 2), and 95 °C for 15 s, 60 °C for 15 s, 65 °C for 5 s, and 95 °C for 5 s (Step 3). Analysis was carried out using the ΔΔC(t) method[58]. Samples were normalized to Gapdh.

*Adra2a*
Forward: CAAGATCAACGACCAGAAGT
Reverse: GTCAAGGCTGATGGCGCACAG
*Arhgap36*
Forward: ACTTAGAGCAGTCCTTGCGG
Reverse: GGTAGAGCTCTGTCCGGCT
*Elavl2*
Forward: GGTACCGCCGCCAGGAAACACAACTGTCTAATGGG
Reverse: GCGGCCGCACTGAGGACAAGAGCTCATTAGGCTTTGT
*Gapdh*
Forward: AGGTCGGTGTGAACGGATTTG
Reverse: TGTAGACAATGTAGTTGAGGTCA
*Igfbp6*
Forward: GGTCTACAGCCCTAAGTGCG
Reverse: AGGGGCCCATCTCACTATCT

*Kctd9*
Forward: CGGGTCACGCTGTTCTTGA
Reverse: ACAGCACATCATCATCCCTGA
*Nefm*
Forward: CAGCTACCAGGACACCATCCAG
Reverse: GTGTACAGAGGCCCGGTGAT
*Peg10A*
Forward: CCGATACACGCGTTTCCAAC
Reverse: TAAAACCCGCCTGTTCCACA
*Peg10B*
Forward: AATCCTCGTGTGGAACAGGC
Reverse: TCATCATCTTCGGCGTCAGG
*Prkcb*
Forward: CAGAGATTGCCATCGGTCTGT
Reverse: CCCCTCAGAATCCAGCATCA
*Scg5*
Forward: ATCAAGGCTACCCAGACCCT
Reverse: GGATTGACACTCCTCCGCTT

**Statistics and reproducibility.** For all experiments, alpha criterion was set to 0.05. SI testing was analyzed for SI ratio and interaction zone time. SI ratio was analyzed by independent samples *t*-tests between groups. Interaction time was analyzed by mixed two-way ANOVAs with Target as the within factor. Omnibus ANOVAs were followed by Holm–Sidak corrected post hoc comparisons between groups. Avoidance learning in the TDPA was analyzed using mixed two-way ANOVAs with Days as the within factor followed by Holm–Sidak corrected post hoc comparisons between groups. EPM and OF behavior was analyzed by independent samples *t*-tests between groups. For novel object recognition experiment, object interaction time was analyzed using mixed two-way ANOVAs with Object as the within factor followed by Holm–Sidak corrected post hoc comparisons within groups. For all western blotting and immunohistochemistry results, data were analyzed by independent samples *t*-tests between groups. Spike number in electrophysiology experiments was analyzed by mixed two-way ANOVAs with Current as the within factor followed by Holm–Sidak corrected post hoc comparisons between groups. *I*–*V* curves in electrophysiology experiments were analyzed by mixed two-way ANOVAs with Voltage as the within factor followed by Holm–Sidak corrected post hoc comparisons between groups. For all other electrophysiological measures: rheobase, spike amplitude, spike half-width, sEPSC amplitude, sEPSC frequency, and other cellular properties (Table S1) data were analyzed by independent samples *t*-tests between groups. Refer to Tables S4 and S5 for all omnibus statistical results.

In most cases, behavioral experiments were conducted in no less than two cohorts to ensure reproducibility. For all viral manipulation experiments, confirmation of viral targeting was conducted using antibodies to enhance native GFP signal (e.g., Fig. 1c). Data from mice lacking targeting in a brain region were removed from analyses. All representative images and electrophysiological recording traces were selected based on data representing the mean for each group. Representative micrograph displayed in Fig. 4e were reproduced in all samples (*n* = 6/group). The same is true in Fig. 5e. FosB staining micrographs displayed in Fig. S1 were replicated; and micrographs and data shown are from a second replication. Representative micrographs in Figs. S4d, S5b, and S8a were replicated in preliminary studies (two replications for S4d; four replications for S5b; one replication for S8a) and reproduced in the final data shown. Representative micrograph shown in Fig. S11b has been replicated multiple times in our lab and published[5]. In addition, all experiments using the methods proposed in Fig. S11a replicated the same pattern of expression shown in S11b.

**Reporting summary.** Further information on research design is available in the Nature Research Reporting Summary linked to this article.

## Data availability

Sequencing datasets generated during and analyzed during the current study are available in the NIH GEO repository (https://www.ncbi.nlm.nih.gov/geo/) with the accession code GSE137283. All the other data supporting the findings of this study are available within the article and its supplementary information files and from the corresponding author upon reasonable request. A reporting summary for this article is available as a Supplementary Information file. Source data are provided with this paper.

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

## Acknowledgements

We would like to thank Kenneth Moon for outstanding technical assistance. This work was supported by NIMH R01 111604, NIDA R01 040621, and NINDS R01 085171 (to A.J.R.) and a NARSAD Young Investigator Award from the Brain and Behavior Research Foundation (to A.L.E.). *Rosa26$^{eGFP-L10a}$* mice were generously provided by Dr. Gina Leinninger at Michigan State University.

## Author contributions

A.L.E., C.E.M., F.M.B., M.M.R., I.M., R.L.N., and A.J.R. conceived and interpreted experiments; A.L.E., C.E.M., E.S.W., R.M.B., P.A.G., A.G., A.J.W., S.A., K.B.-A., and W.E. performed experiments; A.L.E. and A.J.R. wrote the paper; R.L.N. and Y.N.O. contributed tools.

## Competing interests

The authors declare no competing interests.
