## [Peer Review File · Nature Communications]

Reviewers' Comments:

Reviewer #1:

Remarks to the Author:

The manuscript by Eagle et al describes a novel mechanism by which activity of hippocampal afferents to nucleus accumbens is altered by stress and mediates susceptibility to social defeat stress in mice. By employing a combination of intersectional transgenic approaches together with physiological and behavioral experiments, they convincingly show that Δ FosB is a critical cellular element for the circuit-specific function of hippocampal neurons in stress-induced social avoidance.

In this manuscript, the authors are showing that while Δ FosB overexpression reduces excitability in hippocampal-accumbens afferents, a condition that promotes resilience, silencing of the FosB gene at hippocampal-accumbens afferents has the opposite effect, promotes depression.

The experiments performed in this body of work are well designed and timely controlled. Here below you will find some comments to the manuscript:

Major comments:

1- The authors based their interpretations by simply relying on stress-induced social avoidance. I would highly recommend implementing in the current investigation other behavioral tasks that are commonly used in the preclinical depression literature (such as sucrose preference test and forced swim test). By doing so the authors may build a stronger and more generalizable body of data.

2-The authors elegantly define a gene expression profiling at hippocampal-accumbens afferents wherein Δ FosB is capable of repressing the gene coding for α 2AAR. Given that this finding is of incredibly high relevance, I wonder if the authors ever tested in the context of slice preparation how modulating the activity of α 2AAR regulates firing rate of vHPC-NAc in animals that are either resilient or depressed after a social defeat stress paradigm.

3- Moreover, given that the gene coding for α 2AAR is tightly regulated by Δ FosB and may be causally link to depressive phenotypes, then it would be important to experimentally verify that bidirectional control of its expression (overexpression or KO) should phenocopy resilience or susceptibility, respectively. Concurrently, the latter experimental conditions should bidirectionally regulate hippocampal intrinsic excitability.

Reviewer #2:

Remarks to the Author:

Eagle et al. show that circuit-specific Δ FosB in vHPC underlies resilience to stress using multidisciplinary approaches. They also identify α 2AAR as a potential molecular target downstream of Δ FosB. This study provides molecular, cellular and circuit level understanding of stress-induced behavior, which would be of interest to many readers in the broad field of neuroscience. However, there remain a number of questions and concerns that need to be addressed before publication.

Major points:

1) Quantitative comparison of immunohistochemical signals between different slices is often difficult. The staining efficiency differs from slice to slice, and is also different between the surface and deep areas even in the same slice due to the lower penetration of antibodies into deep regions. Thus, Fig. 1a and b are not so convincing. Since the authors already use the TRAP technique to quantify the circuit-specific translation in Fig. 5 in this paper, I recommend the TRAP quantification of Δ FosB expression between control and stress mice to reach the conclusion.

2) Fig. S4 demonstrates that their CRISPR-Cas9-mediated knockout is mild and incomplete. Because they have floxed FosB mice, which they actually use in Fig. 4 and 5 in this paper, it should be much better to use the floxed mice for complete knockout to provide more convincing and straightforward data.

3) In Fig. 2 and 3, the authors "specifically" label vHPC-NAc and vHPC-BLA projections, respectively. However, these projections have collaterals as well, as LeGates et al., Nature 2018 demonstrated the collateral projections of vHPC-NAc to the amygdala. The authors should show how specific their labeling is, to discuss this point in this paper.

4) The authors' model shown in Fig. 5g is overstatement. α 2AAR expression seems to be regulated by Δ FosB, but there is no evidence that α 2AAR plays a role in stress downstream of Δ FosB. To show that, the authors should test if the up- and downregulation of α 2AAR in wild and Δ FosB knockout, respectively, can consistently shift the phenotypes.

Minor points:

5) In Fig. S5b, I wonder why Cas9 is localized not in the nucleus but in the cytoplasm, which looks not suitable for efficient genome editing. Also, did the authors use anti-myc antibody to detect Cas9 in this immunostaining? This should be clarified in the Method section.

6) In Fig. S7a, I wonder why Δ FosB is not detected in GFP-negative cells, while there should be endogenous expression of Δ FosB in a population of cells, as shown in other figures in this paper.

We thank the associate editor and reviewers for their generous comments. We have conducted additional new and replication experiments, clarified our methods, and updated our conclusions based on the reviewers' astute points. We feel that the revised manuscript has significantly improved.

Reviewers' comments:

Reviewer #1 (Remarks to the Author):

The manuscript by Eagle et al describes a novel mechanism by which activity of hippocampal afferents to nucleus accumbens is altered by stress and mediates susceptibility to social defeat stress in mice. By employing a combination of intersectional transgenic approaches together with physiological and behavioral experiments, they convincingly show that Δ FosB is a critical cellular element for the circuit-specific function of hippocampal neurons in stress-induced social avoidance.

In this manuscript, the authors are showing that while Δ FosB overexpression reduces excitability in hippocampal-accumbens afferents, a condition that promotes resilience, silencing of the FosB gene at hippocampal-accumbens afferents has the opposite effect, promotes depression.

The experiments performed in this body of work are well designed and timely controlled. Here below you will find some comments to the manuscript:

Major comments:

1- The authors based their interpretations by simply relying on stress-induced social avoidance. I would highly recommend implementing in the current investigation other behavioral tasks that are commonly used in the preclinical depression literature (such as sucrose preference test and forced swim test). By doing so the authors may build a stronger and more generalizable body of data.

We agree with the reviewer that our focus on social avoidance limits the broad interpretability of the findings. The CSDS model has been characterized in previous literature to induce symptoms besides social avoidance (see Table 1 in Krishnan et al., 2007, *Cell*), but we did not measure these here, and we understand that interpreting social avoidance as “depression” is inappropriate. We have substantially altered the text (including the title) to make the interpretations of differences in social avoidance more direct and to reduce broader interpretations on other preclinical depressive-like symptoms. We also share the reviewer's interest on the role of the vHPC-NAc circuit in other stress-induced preclinical behaviors, however, we feel that these data are outside the scope of the current study, which directly addresses a potential mechanism for this circuit's previously identified role in social avoidance (from Bagot et al., 2015, *Nat Comm*).

2-The authors elegantly define a gene expression profiling at hippocampal-accumbens afferents wherein Δ FosB is capable of repressing the gene coding for α 2AAR. Given that this finding is of incredibly high relevance, I wonder if the authors ever tested in the context of slice preparation how modulating the activity of α 2AAR regulates firing rate of vHPC-NAc in animals that are either resilient or depressed after a social defeat stress paradigm.

We agree with the reviewer that $\alpha 2AAR$ may be an exciting target to regulate excitability in this circuit (as well as a social withdrawal phenotype) and we made novel viral vectors and performed a series of studies that address a direct link between $\Delta FosB$, $\alpha 2AAR$, excitability, and social avoidance would be an interesting story. Unfortunately, all our results were negative. Therefore, we have modified the current manuscript to highlight our approach to identifying gene targets, with *Adra2a* being a validation of the TRAP technique, but not a mechanism for social avoidance. Specifically, we have modified Fig. 5, along with textual changes so that we are not overstating our findings.

3- Moreover, given that the gene coding for $\alpha 2AAR$ is tightly regulated by $\Delta FosB$ and may be causally link to depressive phenotypes, then it would be important to experimentally verify that bidirectional control of its expression (overexpression or KO) should phenocopy resilience or susceptibility, respectively. Concurrently, the latter experimental conditions should bidirectionally regulate hippocampal intrinsic excitability.

See previous answer for our response.

Reviewer #2 (Remarks to the Author):

Eagle et al. show that circuit-specific $\Delta FosB$ in vHPC underlies resilience to stress using multidisciplinary approaches. They also identify $\alpha 2AAR$ as a potential molecular target downstream of $\Delta FosB$. This study provides molecular, cellular and circuit level understanding of stress-induced behavior, which would be of interest to many readers in the broad field of neuroscience. However, there remain a number of questions and concerns that need to be addressed before publication.

Major points:

1) Quantitative comparison of immunohistochemical signals between different slices is often difficult. The staining efficiency differs from slice to slice, and is also different between the surface and deep areas even in the same slice due to the lower penetration of antibodies into deep regions. Thus, Fig. 1a and b are not so convincing. Since the authors already use the TRAP technique to quantify the circuit-specific translation in Fig. 5 in this paper, I recommend the TRAP quantification of $\Delta FosB$ expression between control and stress mice to reach the conclusion.

Response: We agree with the reviewer that a more quantitative validation of CSDS-induced $\Delta FosB$ in this circuit was necessary. We have performed circuit-specific TRAP on pooled samples (n=4-5 per condition) and purified enough mRNA for qPCR. We found a 13-fold increase in $\Delta FosB$ mRNA in this circuit after stress, which aligns with our immunohistochemistry results. The results were added to Figure S1 and we have updated the manuscript to indicate this finding.

2) Fig. S4 demonstrates that their CRISPR-Cas9-mediated knockout is mild and incomplete. Because they have floxed FosB mice, which they actually use in Fig. 4 and 5 in this paper, it

should be much better to use the floxed mice for complete knockout to provide more convincing and straightforward data.

The floxed FosB model does not allow for *circuit specificity* in gene manipulation, which is the reason for our pioneering the novel intersecting CRISPR approach. If we were to simply inject retrograde Cre virus into NAc of floxed FosB mice, all neurons projecting to NAc, including neurons in PFC, amygdala, VTA, and other regions, would lose FosB expression, not just vHPC neurons. With our novel system, we are the first to perform circuit-specific genetic manipulation, allowing us to uncover the novel circuit-specific behavioral effects reported here.

3) In Fig. 2 and 3, the authors “specifically” label vHPC-NAc and vHPC-BLA projections, respectively. However, these projections have collaterals as well, as LeGates et al., Nature 2018 demonstrated the collateral projections of vHPC-NAc to the amygdala. The authors should show how specific their labeling is, to discuss this point in this paper.

The reviewer makes a very important point. We agree that some of these projections may be collaterals, however it is still uncertain how many of the vHPC-NAc population send collaterals to amygdala or any other target region. LeGates demonstrated the existence of collateral projections by examining axonal staining in the target region and only in a small sample. While their data strongly suggested the presence of collaterals it did not indicate the numbers or percentages of collaterals between these two target regions. In order to provide these data, we have conducted immunohistochemistry with dual retrograde injections. We found that the amount of collateral neurons (extending to both nucleus accumbens and amygdala) is scarce. We only found one collateral to both NAc and BLA across 2 mice. The manuscript has been updated to include representative 10x pictures in Fig. S7. However, this could be due to many factors including viral targeting, etc.. In addition, this does not address the collaterals to other regions, e.g. cortex. Therefore, more studies are clearly needed to address this question in full. In addition, the clear difference in behavioral phenotypes when FosB is knocked out of the separate circuits suggests that these circuits are segregated. We have modified the text of the manuscript to suggest the population of collaterals (between these 2 regions) is small, but we underscore the need for a broader study to fully address this question.

4) The authors’ model shown in Fig. 5g is overstatement. $\alpha 2AAR$ expression seems to be regulated by $\Delta FosB$, but there is no evidence that $\alpha 2AAR$ plays a role in stress downstream of $\Delta FosB$. To show that, the authors should test if the up- and downregulation of $\alpha 2AAR$ in wild and $\Delta FosB$ knockout, respectively, can consistently shift the phenotypes.

We agree with the reviewer, and please see our response to Reviewer 1. We have modified the model in the figure 5 and our text in the manuscript to highlight this.

Minor points:

5) In Fig. S5b, I wonder why Cas9 is localized not in the nucleus but in the cytoplasm, which looks not suitable for efficient genome editing. Also, did the authors use anti-myc antibody to detect Cas9 in this immunostaining? This should be clarified in the Method section.

We used a Cas9 antibody for this experiment from Diagenode (methods are now updated with supplier, concentration, and catalog number). As the reviewer points out, the vast majority of Cas9 signal appears to be in the cytoplasm. We have observed this in the supplier's application in Hela cells and in our laboratory. This may be due to the requirement of guide RNA to translocate the Cas9 to the nucleus or because viral overexpression is such that some Cas9 remains in the cytoplasm. In either case, our scrambled gRNA controls demonstrate that the behavioral outcomes are not due to off-target effects of Cas9, in or out of the nucleus.

6) In Fig. S7a, I wonder why Δ FosB is not detected in GFP-negative cells, while there should be endogenous expression of Δ FosB in a population of cells, as shown in other figures in this paper.

We agree with the reviewer that these pictures were not representative – they were underexposed. We replicated the methods from Fig S7a and have included new pictures that address this comment. We have also included 40X representative pictures.

** See Nature Research's author and referees' website at www.nature.com/authors for information about policies, services and author benefits

This email has been sent through the Springer Nature Tracking System NY-610A-NPG&MTS

Confidentiality Statement:

This e-mail is confidential and subject to copyright. Any unauthorised use or disclosure of its contents is prohibited. If you have received this email in error please notify our Manuscript Tracking System Helpdesk team at <http://platformsupport.nature.com>.

Details of the confidentiality and pre-publicity policy may be found here <http://www.nature.com/authors/policies/confidentiality.html>

Privacy Policy | Update Profile

DISCLAIMER: This e-mail is confidential and should not be used by anyone who is not the original intended recipient. If you have received this e-mail in error please inform the sender and delete it from your mailbox or any other storage mechanism. Springer Nature America, Inc. does not accept liability for any statements made which are clearly the sender's own and not expressly made on behalf of Springer Nature America, Inc. or one of their agents. Please note that neither Springer Nature America, Inc. or any of its agents accept any responsibility for viruses that may be contained in this e-mail or its attachments and it is your responsibility to scan the e-mail and attachments (if any).

REVIEWERS' COMMENTS:

Reviewer #1 (Remarks to the Author):

I am happy with the revisions made by the authors. I have no additional comments.

Reviewer #2 (Remarks to the Author):

The authors addressed my concerns, and I think the manuscript is now suitable for publication.